# Protection of Teleost Fish against Infectious Diseases through Oral Administration of Vaccines: Update 2021

**DOI:** 10.3390/ijms222010932

**Published:** 2021-10-10

**Authors:** Jarl Bøgwald, Roy A. Dalmo

**Affiliations:** Norwegian College of Fishery Science, Faculty of Biosciences, Fisheries and Economics, UiT—The Arctic University of Norway, Muninbakken 21, N-9019 Tromsø, Norway; jarl.bogwald@uit.no

**Keywords:** oral vaccination, fish, teleost, mucosal immunity

## Abstract

Immersion and intraperitoneal injection are the two most common methods used for the vaccination of fish. Because both methods require that fish are handled and thereby stressed, oral administration of vaccines as feed supplements is desirable. In addition, in terms of revaccination (boosting) of adult fish held in net pens, oral administration of vaccines is probably the only feasible method to obtain proper protection against diseases over long periods of time. Oral vaccination is considered a suitable method for mass immunization of large and stress-sensitive fish populations. Moreover, oral vaccines may preferably induce mucosal immunity, which is especially important to fish. Experimental oral vaccine formulations include both non-encapsulated and encapsulated antigens, viruses and bacteria. To develop an effective oral vaccine, the desired antigens must be protected against the harsh environments in the stomach and gut so they can remain intact when they reach the lower gut/intestine where they normally are absorbed and transported to immune cells. The most commonly used encapsulation method is the use of alginate microspheres that can effectively deliver vaccines to the intestine without degradation. Other encapsulation methods include chitosan encapsulation, poly D,L-lactide-co-glycolic acid and liposome encapsulation. Only a few commercial oral vaccines are available on the market, including those against infectious pancreatic necrosis virus (IPNV), Spring viremia carp virus (SVCV), infectious salmon anaemia virus (ISAV) and *Piscirickettsia salmonis*. This review highlights recent developments of oral vaccination in teleost fish.

## 1. Introduction

It is suggested that oral immunization evokes a mucosal response, facilitated by the production and secretion of mucosal immunoglobulins (Igs)—where IgT (IgZ in zebrafish (*Danio rerio*)) is central [1,2,3]. The produced IgT may act as arresting molecules (opsonins) that facilitate further degradation of or controlling the microbes [4]. The amount of Igs in normal serum differs between fish species, where, for example, Atlantic salmon (*Salmo salar*) serum contains approximately 1 mg mL^−1^ Igs, whereas haddock (*Melanogrammus aeglefinus*) and cod (*Gadus morhua*) serum have approximately 7 and 12 mg mL^−1^ Igs, respectively. The latter fish species lack genes for MHC class II—which suggests that these two species rely much on “natural” antibodies for protection [5]. The level of Igs in cod serum is quite similar to those found in ray-finned fish [6]. In a recent study, the level of IgT in serum of rainbow trout (*Oncorhynchus mykiss*) was calculated to be more than 600× lower (8.62 µg mL^−1^ in serum) than for IgM, whereas in the nasal mucus, it was approximately 164× lower (1.7 µg mL^−1^) [7]. To our knowledge, the concentration of gut mucus IgM and IgT following oral vaccination has not been evaluated before. Based on present findings, one can suggest that IgT indeed is a mucosal Ig, but it remains to be shown whether IgT is a mucosal correlate of protection. Nevertheless, oral vaccination is an attractive modality to immunize fish for increased protection against pathogens. Since the last review article on oral vaccination by Mutuloki et al. (2015) [1], there have been several advances, which are highlighted in the current overview together with previously published data.

## 2. Bacterial Diseases

### 2.1. Vibriosis

Vibriosis is one of the most severe infectious diseases affecting marine fish in tropical and subtropical zones of the world. Its pathogens mainly include *Vibrio harveyi*, *Vibrio parahaemolyticus*, *Vibrio alginolyticus*, and *Vibrio anguillarum*. Mao et al. (2011) used outer membrane protein K (OmpK), one of the significant outer-membrane proteins of *V. harveyi*, as an oral vaccine [8]. The molecule was expressed in the yeast *Pichia pastoris* and encapsulated in alginate microspheres. The vaccine was fed to Japanese sea bass (*Lateolabrax japonicus*). A significant antibody level against OmpK was elicited and the challenge of vaccinated fish revealed protection against the pathogen. *V. anguillarum* is the main causative agent of vibriosis in cultured sea bass. Galindo-Villegas et al. (2013) used a commercial sea bass oral vaccine against *V. anguillarum* and improved it with recombinant sea bass tumor necrosis factor alpha (rTNFα) as an adjuvant [9]. Orally immunized European sea bass (*Dicentrarchus labrax*) conferred protection against *V. anguillarum* challenge throughout a short time period. Moreover, fish that received adjuvant + antigen significantly extended the response. In both cases, achieved protection was independent of serum IgM. However, IgT transcripts were found to increase in the gut of rTNFα-treated fish. Fish treated with rTNFα also showed a dramatic change in their T lymphocytes distribution and localization in the gut mucosal tissue. In another study by Sarropoulou et al. (2012), a commercial oral vaccine against *V. anguillarum* (Aquavac Vibrio Oral, ISPAH) was fed for 10 days to European sea bass and boosted four months later [10]. The evaluation revealed specific immune gene expression profiles in the gut.

Turbot (*Scophthalmus maximus*) is one of the most important commercial fish species worldwide because of its rapid growth and high market value. *V. anguillarum* is causing serious infections in this species. Gao et al. (2016) developed carboxymethyl chitosan/chitosan nanoparticles (CMCS/CS-NPs) loaded with extracellular products (ECPs) of *V. anguillarum* for oral vaccination. Immunized fish showed elevated specific antibody levels and higher concentrations of lysozyme and complement activities compared with stimulation with soluble ECP. A DNA vaccine was constructed using the *ompK* gene of *V. parahaemolyticus* encapsulated in chitosan nanoparticles for oral delivery to black seabream, *Acathopagrus schlegelii* Bleeker [11]. The *ompK* gene transcripts were found in the mid-intestine, liver, kidney and muscle after administration, and the expression evoked an immune response that protected fish against infection (RPS of 72.3%). In a study by Li et al. (2016) goldlined/silver sea bream juveniles (*Sparus sarba*) were fed with pellets containing a vaccine of formalin-killed *V. alginolyticus* [12]. Three weeks after booster vaccination, fish were challenged with pathogenic *V. alginolyticus*. Although oral vaccination could not offer protection as good as that of injection vaccination, oral vaccination appeared effective for protecting silver sea bream [12].

### 2.2. Motile Aeromonad Septicemia (MAS)

*Aeromonas hydrophila* is a Gram-negative, facultative anaerobic motile bacterium that is the causative agent of motile Aeromonas septicemia (MAS) in fish. The South American catfish, the hybrid surubim (*Pseudoplatystoma corruscans* × *Pseudoplatystoma reticulatum*), has a high economic value because of its excellent flavor and spineless fillets. *A. hydrophila* is a serious pathogen of this species. Do vale Pereira et al. (2015) determined the efficacy of different inactivated *A. hydrophila* vaccines administered intraperitoneally with or without an oral booster [13]. Fish treated both intraperitoneally and orally boosted with bacterin + toxoid (inactivated extracellular products) showed the lowest cumulative mortality (10%) when intraperitoneally challenged with *A. hydrophila*. Dubey et al. (2016) used the cloned outer membrane protein W (OmpW) of *A. hydrophila* as a vaccine component in rohu (*Labeo rohita* Hamilton) and encapsulated it in poly D,L-lactide-co-glycolic acid (PLGA) nanoparticles for oral vaccination [14]. Two antigen doses were orally administered. The high antigen dose resulted in higher antibody levels than the low antigen dose after oral vaccination and corresponded with the relative percent survival (RPS). 

### 2.3. Edwardsiellosis

Chatakondi et al. (2018) evaluated the efficacy of an oral live-attenuated *Edwardsiella ictaluri* vaccine against enteric septicemia of catfish (ESC) in channel (*Ictalurus punctatus*) and hybrid catfish (*Ictalurus punctatus × Ictalurus fucatus*) [15]. Channel catfish is the single largest aquacultured species in the USA. Enteric septicemia of catfish (ESC) and columnaris disease are two of the most economically important bacterial diseases affecting the catfish industry. Thirty-five days post-vaccination, fish were challenged with virulent *E. ictaluri* and mortality was examined for 30 days post-challenge. The mortality of non-vaccinated hybrids (85%) and non-vaccinated channel catfish (75%) was significantly greater than all groups of vaccinated. In channel catfish, mortality in the lowest dose of vaccine was 26.6% (RPS 61.9%) and significantly higher than in the highest dose group (14.1%, RPS 80.6%). Mortality of orally vaccinated hybrid catfish ranged between 10.4%, (RPS 87.4%) and 14% (RPS 83,5%) [15]. Recombinant outer membrane protein A (rOmpA) of *Edwardsiella tarda* was encapsulated in chitosan polymeric nanoparticles (NPs) and used for oral vaccination of fringed-lipped peninsula carp (*Labeo fimbriatus*) [16]. All vaccines were administered in the feed for 21 days. The fish were challenged with a pathogenic strain of *E. tarda* 51 days post-vaccination. The NP-rOmpA vaccine produced higher antibody levels and had superior protection than the inactivated whole cell vaccine of *E. tarda*. Kole et al. (2018) report of a bicistronic DNA vaccine containing glyceraldehyde-3-phosphate dehydrogenase gene (*gapdh*) of *E. tarda* with an additional immune adjuvant gene (interferon gamma gene (*ifng*)) (*gapdh + ifng*) nanoconjugated with chitosan nanoparticles [17]. The vaccine was given to rohu fingerlings in the feed for 14 days, and a RPS of 81,82 % was obtained. This group also showed significant upregulation of immune gene transcripts like *ighc* (IgM heavy-chain constant-region), *inos*, *tlr22*, *nod1* and *il1b*.

### 2.4. Yersiniosis

*Yersinia ruckeri*, a Gram-negative member of the family Enterobacteriaceae, is the causative agent of enteric redmouth disease (ERM) and yersiniosis in salmonids. Ghosh et al. (2016) used a microencapsulated *Y. ruckeri* vaccine formulation for both immersion and oral administration to first-feeding Atlantic salmon [18]. Both oral and oral + dip groups demonstrated moderate protection when challenged (RPS of 29.4% and 51%, respectively). Protection was even lower for fish dipped (immersion vaccinated) one and two times (RPS 20.4% and 16.7%, respectively) which were immunized only via immersion. 

### 2.5. Lactococcosis/Streptococcosis

The Gram-positive bacteria, *Streptococcus iniae* and *Lactococcus garviae*, are known as two major species of streptococcal infections in many fish species, particularly salmonids. Romalde et al. (2004) studied both non-encapsulated and alginate encapsulated oral vaccines against lactococcosis in rainbow trout. The encapsulated vaccine administered orally resulted in a RPS of 50% [19]. The same vaccine was used as a booster immunization after a priming intraperitoneal vaccination. Halimi et al. (2018) used Eudragit L30D-55-coated vaccine (polymer-drug release coating containing formalin-inactivated bacteria) pellets for vaccination of rainbow trout against *S. iniae* and *L. garviae* [20]. Vaccination was conducted for 14 days, and 60 days thereafter, the fish were challenged with either *L. garviae* or *S. iniae*. Observed survival percentages were 72% (*L. garviae*) and 85% (*S. iniae*). Another fish species that have been shown to suffer from streptococcosis is tilapia. Red hybrid tilapia (*Oreochromis* sp.) was immunized by an inactivated recombinant vaccine expressing the cell wall surface anchor family protein of *Streptococcus agalactiae*. Orally immunized fish developed a strong and significantly high IgM antibody response in serum, mucus and gut lavage fluid samples. The rate of survivors was 70% [21]. A feed-based oral vaccine of formalin-fixed *S. agalactiae* and in addition a vaccine containing Freund’s incomplete adjuvant were prepared [22]. Red tilapia were fed the respective vaccines and boosted orally at day 14. Four weeks after the start of the experiment, fish were intraperitoneally challenged with *S. agalactiae*. The adjuvanted group showed a significantly higher survival rate. In a study by Hayat et al. (2021), red hybrid tilapia were fed a formalin-killed oral *S. iniae* vaccine [23]. All vaccinated groups showed a significant increase in anti-*S. iniae* IgM levels in serum, mucus and gut-lavage. Enhanced protection was achieved after a booster dose. The study of Monir et al. (2020) focuses on the effectiveness of a newly developed feed-based killed bivalent vaccine against *S. iniae* and *A. hydrophila* in hybrid tilapia (*Oreochromis mossambicus* × *O. niloticus*) [24]. The vaccine was orally administered on days 0, 14 and 42 days. Challenge tests were performed 70 days post-vaccination with single bacterial infections and co-infections. Results showed a significant protection after the coinfection challenge.

### 2.6. Furunculosis

Furunculosis is a highly contagious disease that affects fish of all ages. The infection causes high mortality in salmonids and some other species of fish like the goldfish (*Carassius auratus*). Maurice et al. (2004) cloned and modified the A-layer protein of atypical *Aeromonas salmonicida* by genetic fusion of the protein transduction protein (MTS) derived from Kaposi fibroblast growth factor (At-MTS) [25]. Both At-R (A-layer) and At-MTS were encapsulated in alginate beads and fed to goldfish. Vaccine-fed goldfish demonstrated increased antibody titers. In spite of this, vaccinated fish did not demonstrate resistance to infection with atypical *A. salmonicida*. Furunculosis caused by *A. salmonicida* ssp. *salmonicida* is one of the most important bacterial diseases affecting cultured and wild salmonid fish, and also in non-salmonid marine species such as turbot (*S. maximus*), gilthead sea bream (*Sparus auratus*) and sea lamprey (*Petromyzon marinus*). A commercial furunculosis vaccine induced protection in turbot after injection, with RPS-values of 72–99% 120 days post-immunization [26]. Six months after immunization the RPS levels declined to 50–52%. An oral boost after the primary injection vaccination did not enhance protection. The response of isolated gut leucocytes to four bacterins of *A. salmonicida* prepared from different strains, combinations and strains grown in different environments in comparison to a *Y. ruckeri* bacterin was studied by Attaya et al. (2020) [27]. A wide array of immune gene transcripts was modulated by the bacterins. The gut-associated lymphoid tissue (GALT) leucocyte responses were sensitive enough to distinguish the different bacterial species, strains and membrane proteins.

### 2.7. Francisellosis

Members of the genus *Francisella* are small Gram-negative facultative intracellular bacteria that cause francisellosis in a wide variety of fish species worldwide. *Francisella noatunensis* subsp. *orientalis* (Fno) has been recently described as a warm-water pathogen of tilapia *Oreochromis* spp. Hoare et al. (2021) used an existing bacterin to optimize a mucosal vaccine against Fno in Nile tilapia (*Oreochromis niloticus*) with mucosal adjuvants (IMS 1312 VG PR; a Montanide™ variant) [28]. Fish fry received the vaccine with oral gavage and were boosted once. Specific IgM titers were significantly elevated in serum and mucus in fish given the highest dose.

### 2.8. Piscirickettsiosis/Salmonid Rickettsial Septicaemia

One of the main pathogens that plagues salmonid culture during the on-growing phase is *Piscirickettsia salmonis*, the causal agent of salmonid rickettsial septicaemia (SRS) or piscirickettsiosis. This Gram-negative bacterium, fastidious intracellular pathogen, originally isolated from a coho salmon (*Oncorhynchus kisutch*) in southern Chile, produces a systemic infection characterized by colonization of several organs, including kidney, liver, spleen, intestine, brain, ovary and gills. SRS is the infectious disease that produces the highest losses in the Chilean salmon industry. The effect of alginate-encapsulated *P. salmonis* antigens (AEPSA) incorporated in the feed as an oral vaccine in Atlantic salmon (Providean Aquatech 1, Anasac, Chile) was studied by Sotomayor-Gerding et al. (2020) [29]. The oral vaccine produced an acquired immune response (IgM) similar to the injectable vaccine at 840 days degree. Tobar et al. (2015) studied the vaccination program against *P. salmonis* and infectious salmon anemia virus to maintain long-term protection in farmed salmonids in 622 Chilean farms from 2011–2014 [30]. All fish were vaccinated first by the intraperitoneal injection of oil-adjuvanted antigen and then by oral vaccines as a booster vaccination. A close association between antibody levels and protective status was found. The results of this study demonstrated that, in the field, several oral immunizations are essential to uphold a high level of specific anti-pathogen antibodies and the protective status during the whole production cycle [30]. 

An overview of oral vaccines against bacterial diseases is found in Table 1.

## 3. Viral Diseases

### 3.1. Infectious Pancreas Necrosis (IPN)

IPN is a highly infectious viral disease, causing high mortalities in fish species worldwide. The IPNV belongs to the family Birnaviridae, genus Aquabirnavirus with a bisegmented double-stranded RNA genome. Segment A is encoding the two major structural proteins of the virus (VP2 and VP3). VP2 is a protective antigen that induces virus neutralizing antibodies. Ahmadivand et al. (2017) developed a DNA vaccine encoding the VP2 gene of IPNV and encapsulated it in chitosan/tripolyphosphate (CS-TPP) nanoparticles and alginate microparticles [43]. Rainbow trout fry (weight 3 g) were orally immunized with feed pellets containing the nanoparticles or microparticles for 90 days. 45 days after vaccination start the fish were challenged by intraperitoneal injection of a virulent isolate of the IPNV. Fish vaccinated with alginate microparticles containing 10 and 25 mg of the VP2 DNA vaccine gave cumulative mortalities of 24% and 10% resulting in RPS of 59% and 82% respectively. Cumulative mortalities in the CS-TPP-groups were 30% and 17% with corresponding RPS of 47% and 70 % for the 10 and 25 mg DNA vaccine groups, respectively [43]. De las Heras et al. (2010) found that a VP2 capsid gene encapsulated in alginate microcapsules gave RPS of 80% when administered orally to brown trout (*Salmo trutta*) when challenged with IPNV 15 and 30 days after vaccine delivery [44]. Ballesteros et al. (2012) showed that oral VP2 vaccine could mimic both the time-course and the organ (head kidney, spleen, intestine, pyloric ceca, thymus) profiles of transcripts obtained after IPNV infection [45], but transcriptional levels were lower in VP2-vaccinated than in IPNV-vaccinated trout. They suggested that the mechanisms by which alginate-oral DNA-vaccination induces protection were similar to the defense mechanisms induced by IPNV-infection. In response to oral vaccination with the VP2 DNA vaccine encapsulated in alginate, the B cell response in five different segments of the rainbow trout digestive tract was studied [46]. The pyloric ceca region was the area in which the major recruitment of B cells was demonstrated both IgM^+^ and IgT^+^ intraepithelial lymphocytes [46]. Boosting may be a good method for enhancing or extending protection.

One year after intraperitoneal injection with an oil-adjuvanted IPN-vaccine, Atlantic salmon post-smolts were orally boosted by alginate-encapsulated IPNV antigen (ENCAP) or soluble antigen [47]. This was done twice, seven weeks apart. Compared to controls, fish fed with ENCAP had a significant increase in serum antibodies following the first boost, but not after the second. Results of this study suggest that parenteral prime vaccination with oil-adjuvanted vaccines and followed by oral boost with ENCAP gives an augmentation of the systemic immune response. Symmetrical prime and boost oral vaccination with ENCAP results in augmentation of mucosal immune response. Symmetrical priming and boosting orally with soluble antigens resulted in the induction of systemic immune tolerance [47]. A liposomal DNA vaccine coding for the immunogenic region of the IPNV VP2 capsid protein (lipoplex) induced detectable levels of VP2-specific antibodies and conferred significant protection following IPNV challenge with RPS of 58.2% for single-dose and 66% for double-dose orally administration in Atlantic salmon [48]. A DNA vaccine based on the VP2 gene was given orally to rainbow trout through the feed. A strong protection was obtained significantly upregulating *ifn1*, *ifng*, *mx1*, *il8* (*cxcl8*), *il12*, *igm* (constant region of the heavy chain) and *igt* (constant region of the heavy chain) [49].

### 3.2. Viral Nervous Necrosis

Viral nervous necrosis (VNN), also known as viral encephalopathy and retinopathy (VER), is one of the most serious infectious diseases affecting marine aquaculture. Since its first description in the late 1980s, many economically relevant fish species has proven to be susceptible to the disease, such as European sea bass, Asian sea bass (*Lates calcarifer*), groupers (*Ephinephelus* spp.), striped jack (*Pseudocaranx dentex*), Senegalese sole (*Solea senegalensis*), turbot, Atlantic halibut (*Hippoglossus hippoglossus*) and Atlantic cod. Chen et al. (2011) focused on the development of a new oral vaccine that could provide sufficient, earlier protection in the orange-spotted grouper (*Epinephelus coioides*) [50]. *V. anguillarum* was used as an antigen expression host due to its immune-stimulatory capability. An expression vector containing *heat shock protein 60* gene (*hspd*) (named *groE* in orange-spotted grouper) as an inducible promotor was constructed to express recombinant NNV coat protein. The recombinant *V. anguillarum* was inactivated and encapsulated in *Artemia* nauplii to formulate an oral vaccine [51]. A higher specific antibody titer against NNV was observed in the first week after inoculation [50]. In addition, a higher survival rate compared to an equal dose of an *E. coli*-based oral vaccine was found. Lin et al. (2007) did a similar study in grouper larvae with a recombinant NNV-capsid protein. After oral administration, the vaccine was absorbed in the hindgut, induced anti NNV VP specific antibodies and resulted in a RPS of 64.2–69.5% [52].

Kai et al. (2014) performed an interesting study on immune gene expression in grouper larvae after immunization with binary ethylenimine inactivated NNV [53]. Both bath and oral vaccination triggered gene expression of both humoral and cellular immunity. The red-spotted grouper nervous necrosis virus (RGNNV) is the most common species of NNV worldwide. Cho et al. (2017) developed a recombinant *Saccharomyces cerevisiae* RGNNV capsid protein [54] as an oral vaccine in convict grouper (*Hyporthodus septemfasciatus*) [55]. The oral vaccinated fish produced serum RGNNV neutralizing antibody titers >10^3^ sustained for at least 95 days post-immunization. In response to a challenge with RGNNV, the fish showed significantly reduced mortality and had reduced brain RGNNV titers [55]. Furthermore, an experimental oral vaccine was made from recombinant orange-spotted NNV capsid protein produced in *Escherichia coli* and then assembled into virus-like particles (VLPs). After executive feeding of the VLPs and experimental challenge, the RPS was calculated to be 52.3% [56]. In another study, *RNA2* capsid protein gene of NVV was encapsulated in chitosan-tripolyphosphate nanoparticles and given by feeding. After challenging the Asian sea bass with intramuscular injection of NVV, the orally vaccinated fish exhibited a RPS of 60% [57].

Cho et al. (2018) expressed NNV coat protein in tobacco chloroplasts and used it as an oral vaccine in seven-band grouper (*Epinephelus septemfasciatus*). Fish developed significantly higher antibody titers against the NNV coat protein and were partially protected against viral challenge [58]. An oral vaccine against VNN supplemented with capsaicin conferred protective immunity on seven-band grouper fed for seven days and challenged at 21 days after the start of immunization [59]. Valero et al. (2016) used an oral chitosan-encapsulated DNA vaccine (CP-pNNV) for the nodavirus to protect European sea bass against VNN [60]. The vaccine failed to induce serum circulating or neutralizing specific antibodies (IgM). However, the vaccine upregulated the expression of genes related to cell-mediated toxicity (CMC) and the interferon pathway. In addition, three months after vaccination, challenged fish showed a retarded onset of fish death and lower cumulative mortality with a RPS rate of 45% [60].

### 3.3. Grass Carp Reovirus (GCRV)

Haemorrhagic disease caused by grass carp (*Ctenopharyngodon idella*) reovirus (GCRV) results in a tremendous loss in the grass carp industry. GCRV is a double-stranded RNA virus belonging to the genus Aquareovirus, family Reoviridae. The Gram-positive soil bacterium *Bacillus subtilis* is a non-pathogen, and its spore form is currently used as a probiotic for both human and animal consumption. *B. subtilis* spores have been shown to be able to protect surface-displayed heterologous antigens against degradation. Two grass carp-sourced *B. subtilis* spore-based vaccines (*GC5-VP4* and *GC5-NS38*) were constructed and their potential as oral candidate vaccines against GCRV II was investigated [61]. Both VP4 and NS38 proteins of GCRV II were efficiently displayed on the spore surface of *B. subtilis* GC5. After oral administration, both vaccines increased the survival rate of grass carp against GCRV II with relative percent survival rates of 30% and 36.4%, respectively.

Another recent study by Sun et al. (2020) of the same species used Cot B and Cot C coat anchors for grass carp reovirus recombinant VP7 antigen [62]. In fish orally vaccinated with the VP7-expressing spores, immunogenicity and protection were augmented. The IgM titers were significantly increased on day 7 and reached peaks on day 14, then dropped from day 21 to day 28.

Another oral subunit vaccine against GCRV was constructed for the same species [63]. Outer capsid proteins VP5 and VP7 expressed in *E. coli* were coated on pellets and fed to grass carps for 21 days. Specific antibody responses could be detected in sera and significantly lower accumulative post-vaccination mortality rates were achieved.

### 3.4. Spring Viremia of Carp (SVC)

Spring viremia of carp virus (SVCV) characterized as a member of the genus Sprivivirus of the family Rhabdoviridae is a linear single-stranded negative RNA virus causing high mortality and infectious disease accompanied with typically acute hemorrhage symptoms in cyprinids, in particular common carp. Cui et al. (2015) developed a genetically *Lactobacillus plantarum* co-expressing glycoprotein (G) of SVCV and ORF81 protein of koi herpesvirus (KHV) [64]. The recombinant plasmid was electroporated into *L. plantarum*. Common carps (*Cyprinus carpio*) and koi carps (*Cyprinus carpio koi*) were orally immunized with *L. plantarum* containing the expressed proteins. This vaccination strategy induced significant levels of IgM and reduced viral loads after viral challenge. Also, an effective protection rate of 71% in vaccinated carps and 53% in vaccinated koi until days 65 post-challenge was obtained [64]. A chitosan-alginate microcapsule probiotic *L. plantarum* expressing G protein of SVCV was constructed [65]. Common carps were primarily vaccinated by oral administration and after 14 days a booster vaccine was given and after 28 days, a second booster vaccine was administered. A significantly higher level of antigen-specific IgM antibodies was elicited and the vaccine provided effective protection against SVCV infection 42 days after the primary vaccination [65].

### 3.5. Viral Haemorrhagic Septicaemia

Viral haemorrhagic septicemia (VHS) afflicts over 50 species of freshwater and marine fish of the Northern Hemisphere. The viral haemorrhagic septicaemia virus (VHSV) is an enveloped ribonucleic acid (RNA) virus that belongs to the genus *Novirhabdovirus* of the family Rhabdoviridae. Different strains of the virus occur in different regions and affect different species. There are no signs that the disease affects human health. VHS is also known as Egtved disease. In a study by Kim et al. (2019), cholera toxin B subunit (CTB)-fused recombinant viral haemorrhagic septicaemia virus glycoproteins (rec-VHSV-GPs) were expressed in tobacco, *Nicotiana benthamiana* and used as a combination vaccine (intraperitoneal and oral) in olive flounder (*Paralichthys olivaceous*) [66]. The antibody titers were significantly increased and after viral challenge fish were protected with a significantly lower mortality rate.

### 3.6. Infectious Hematopoietic Necrosis (IHN)

Infectious hematopoietic necrosis virus (IHNV) is responsible for important losses in the salmonid farming industry worldwide. IHNV is a non-segmented, enveloped, single-stranded, negative-sense RNA virus belonging to the genus Novirhabdovirus in the family Rhabdoviridae. Previous studies demonstrated that the IHNV G protein is the only viral protein capable of inducing a neutralizing antibody response to IHNV. In a IHNV vaccination study, a DNA vaccine encoding the IHNV G protein was encapsulated into alginate microspheres and orally administered to rainbow trout. IHNV G transcripts were detected in gills, spleen, kidney and intestinal tissues of vaccinated fish [67]. The oral route was found to require approximately 20-fold more plasmid DNA than the injection route to induce the expression of significant levels of *ihng* transcripts in the kidney and spleen. When vaccinated fish were challenged by immersion with live IHNV, evidence of a dose-response could be observed. The protective effects were partial, but significant differences in cumulative mortalities among the orally vaccinated fish and the unvaccinated or empty plasmid-vaccinated were observed.

### 3.7. Infectious Salmon Anaemia

Infectious salmon anaemia (ISA) is an infectious viral disease of Atlantic salmon. The disease was first reported in Norway in 1984, but has since been reported in Canada, the USA, the Faroe Islands, Ireland and Scotland. Atlantic salmon is the only susceptible species known to develop clinical disease, but ISA virus can replicate in rainbow trout and sea trout. In the following experimental approach, a non-adjuvanted ISAV vaccine (inactivated virus) and a vaccine based on ISAV + plasmid DNA encoding the replicase of alphavirus protein as adjuvant were encapsulated in chitosan nanoparticles—and administered orally to Atlantic salmon [68]. Vaccination with the non-adjuvanted vaccine (inactivated virus) resulted in modest protection, while the adjuvanted gave a RPS of 77%.

### 3.8. Iridoviral Disease (IVD)

Iridovirus is a double-stranded DNA virus of the Iridoviridae family. The main pathogen is the Megalocytivirus. This is further composed of infectious spleen and kidney necrosis virus (ISKNV) and red sea bream iridovirus (RSIV). In the study of Seo et al. (2013), a recombinant major capsid protein (rMCP) of rock bream iridovirus (RBIV) was expressed in yeast. Rock bream (*Oplegnathus fasciatus*) immunized orally with rMCP underwent a successful induction of specific antibodies and was protected against viral challenge [69]. In the study by Shin et al. (2013) a recombinant major capsid protein (rMCP) of rock bream iridovirus was expressed in transgenic rice callus. Rock bream immunized orally with rMCP underwent successful induction of antibodies and was protected against viral challenge [70]. An overview of oral vaccines against viral diseases is found in Table 2.

## 4. Parasitic Diseases

### ClonorChiasis

Since 1998 where Speare et al. [75] showed that rainbow trout orally intubated with gill preparation containing *Loma salmonae* exhibited a certain protection level against reinfection, no major breakthrough of oral vaccines against parasites has occurred. The lack of breakthroughs is also mirrored in the review article by Mutoloki et al. (2015) [1]. However, some recent reports suggest that novel vaccine vehicles may be a solution to induce antiparasitic effects upon oral delivery. Clonorchiasis (*Clonorchis sinensis*) is a serious zoonotic parasitic disease in South Asian regions such as China, Korea, Vietnam and Russia. Approximately 15 million people are infected with this tropical disease globally. Grass carps are the main species of freshwater aquaculture in China with high economic value [76]. People infected with *C. sinensis* are mainly due to eating raw or undercooked freshwater fish containing infective metacercariae, so cutting off the transmission route by interrupting the formation of metacercaria in freshwater fish would be an effective strategy to control clonorchiasis. The Gram-positive *B. subtilis* spores have been shown as an ideal vaccine delivery system and the *C. sinensis* enolase (CsENO) was used as a vaccine candidate. Grass carps were fed spores with CsENO and the fish developed specific IgM levels in serum, intestine and skin mucus in addition to upregulation of central innate and adaptive immunity molecules [76]. In a recent study by Sun et al. (2020b) paramyosin of *C. sinensis* (CsPmy) was expressed on the surface of *B. subtilis* spores [61]. The recombinant spores were incorporated in pellets and administered orally to grass carp. This induced both systemic and local mucosal immune responses and elicited promising protective effects in grass carp. *Ichthyophthirius multifiliis* is a protozoan that invades the gills and skin surfaces of freshwater fish. In the study by Yao et al. (2016) *L. plantarum* NC8 was used as a host to express the mobilization antigens—potential vaccine candidates for prevention of the “white spot” disease (Ichthyophthiriosis). Goldfish were orally immunized with the recombinant vaccine. The antibody level in the blood and skin was increased. In addition, the expression of *igm*, *c3* and *mhc1* genes was significantly upregulated, and after the challenge the average survival rate was elevated. Heidarieh et al. (2015) reported a nanoparticle system for oral delivery of the vaccine. Irradiated *I. multifiilis* (inactivated trophonts) were encapsulated in alginate particles and given orally to rainbow trout. Fish were subsequently challenged with *I. multifiliis*. Differences in hematocrit, red blood cells and biochemical levels (total plasma protein, plasma alkaline phosphatase and plasma glucose levels) were tested—in the form of preclinical assessment. Table 3 gives an overview of oral vaccines against parasites, examined in experimental trials.

## 5. Model Antigens

Kwon et al. (2019) used an oral vaccination platform of microalgae for the delivery of recombinant drugs. The green fluorescent protein (GFP) was bioencapsulated and orally delivered to zebrafish. The GFP was clearly observed in the intestinal tissues and the blood [77]. Plasmid pCMVb (*lacZ* reporter gene encoding for *E. coli* β-galactosidase) was encapsulated in chitosan nanoparticles and orally intubated in gilthead sea bream [78]. The *lacZ* gene expression (β-galactidase activity) could be detected in fish tissues following oral administration for up to 60 days. The results suggest that chitosan nanoparticles enabled efficient oral delivery of pDNA followed by organ migration and expression. In comparison, after parenteral administration, reporter gene expression was mostly restricted to adjacent tissues of the injection site. Organ distribution of the gene expression was more evident after oral administration in the gut, liver and muscle. Moreover, the IgM response was more intense after oral delivery of the plasmid [78]. The findings of Sato & Okamoto (2008) strongly suggest that oral administration of hapten-modified cellular antigens to ginbuna crucian carp (*Carassius auratus langsdorfii*) can induce antigen-specific cytotoxic cells that are capable of recognizing antigens in a MHC-restricted manner, and also of inducing cytotoxic memory response [79]. An overview of oral vaccines using model antigens is found in Table 3.

**Table 3 ijms-22-10932-t003:** Overview of recent oral vaccination trials against a parasitic disease and experiments using model antigens.

Disease	Pathogen	Species	Vaccine	Reference
Clonorchiasis	*Clonorchis sinensis*	Grass carp	Recombinant	[61]
Ichthyophthiriosis	*Ichthyophthirius* *multifiliis*	Goldfish		[80]
		Rainbow trout	Nanoparticle gamma-ray irradiated vaccine	[81]
			Model antigens	
		Gilthead sea bream	Plasmid DNA β-galactosidase (*GLB1*)	[82]
				[78]
		Red crucian carp	OVA model antigen	[83]
		Zebrafish	*GFP* in *Chlamydomonas reinhardtii*	[77]

## 6. Intestinal Immune Defense

### 6.1. Immune Molecules in Mucus and Bile

In a recent study, a proteomic approach was performed to elucidate the constituents of intestinal mucus and bile (tilapia) with a focus on immune-related molecules [84]. The study revealed that the mucus constituted of molecules associated with pattern recognition, antigen presentation, inflammatory cytokines and receptors, adaptors, effectors and signal transduction, T- and B-cell antigen activation and diverse molecules related to immune response. On the other hand, the bile contained proteins related to mainly acute phase response and complement cascade [84]. This suggests that these two different secretions possess specialized functions. We may further suggest that the immune molecules in bile excretions may have a more arresting function (complement) than signaling (and effector) function posed by mucous secretions in the intestine.

### 6.2. Anatomical Distribution of Immune Cells in the Gut

During recent years, many excellent review articles describing the fish’s intestinal immunity have been published [85,86,87,88,89,90,91]. It is acknowledged that the intestine is composed of mainly enterocytes and goblet cells [92,93], with interstitial or associated T- and B-cells [94]. It is not fully clear whether other cell types exist. There are no indications that confined lymphoid structures are present along the gastrointestinal tract of bony fish. However, a lymphoid structure in the cloacal region was recently discovered in Atlantic salmon [95]. This may be the same as Inami et al. (2009) found in Atlantic cod—although the work on cod did not describe the anatomical localization properly [96]. This lymphoid cell aggregate may be active in antigen capture and immune response. Through gene expression studies, genes encoding *il1b*, *cxcl8*, *il10*, *hamp*, and *ccl19* were found. These genes likely play roles at least in the innate immunity. 

An interesting approach to dissect the different cell population in zebrafish intestine was undertaken using single-cell sequencing [97]. With this method, the authors revealed epithelial-like cells with genotypes (claudins, annexins, and endocytosis regulators) similar to mammalian M-cells, and enterocytes expressing gene products that transduce interferon pathway signals. After in-situ hybridization using a probe detecting the M-cell marker *icn* (ictacalcin), it was evident that the M-like cells were localized in the posterior region—analogous to the localization in mammalian colon. It should be noted that no universal M-cell markers have been found [98], thus *icn* could be considered a zebrafish M cell marker. This zebrafish study suggested that there are different cell populations in the fish intestine, though not clearly visible from morphological and anatomical analysis. It has been shown that IgM^+^ and IgT^+^ cells usually have been found in the lamina propria [99], and IgT^+^ cells as intraepithelial cells [100]. However, following oral vaccination using an alginate-encapsuled DNA vaccine (IPNV), IgM and IgT responses were found mainly in the pyloric caeca [46]. The reason for the anatomical redistribution of IgT^+^ and IgM^+^ cells after vaccination is not clear. Obviously, more targeted research should be performed in order to find the exact responding site for antibody response in fish. This applies to detailing the anatomical distribution of different cell populations in many fish species.

### 6.3. Lamina Propria Eosinophilic Granular Cells

In many fish species, eosinophilic granular cells (EGCs) are quite dominant cell populations in the lamina propria (Figure 1), where they reside on both sides of stratum compactum. Since they are numerous, they must possess an important role. It is suggested that fish intestinal eosinophilic granular cells (EGCs) are homologous to mammalian mast cells [101,102,103,104,105]. Rodlet cells, which share many characteristics with EGC, may be precursors of EGC, as suggested by Reite and Evensen (2006) [102]. Whether all fish species possess mast cells analogues with similar functions remains to be shown. Notwithstanding, the number of EGCs increases during intestinal inflammation [102,106,107,108]. Mast cells of perciformes are of special interest because they also contain antimicrobial peptides named “piscidins” [109,110]. Furthermore, zebrafish mast cells have been found to express *myd88* (toll-like receptor adaptor molecule) and *FcRI*, which is similar to IgE receptors in higher vertebrates [111]. 

### 6.4. Modulation of Intestinal Responses

It is clear that the intestine of fishes can respond immunologically. Studies of intestinal mucous factors must be carefully planned and executed to avoid contaminant cells and blood to avoid false assumptions with regard to the actual presence of immune factors in mucous secretion. The same applies to the analysis of intestinal tissue, where most mucous secretions should be removed prior to assessment.

Transcriptomic and proteomic assessment of fish intestines will provide large datasets that can be further refined to study intestinal responses to, e.g., oral vaccination, infection, and probiotics in a more targeted way. From the refined dataset (e.g., differential expressed genes, proteins, correlates of protection or infection), the research community may select central components decisive for given response or/and protection. Recently, there have been many innovative approaches to better understanding intestinal immunity. In one of these studies, proteomic and transcriptomic examination of the intestinal mucus in Tilapia infected with *S. agalactiae* showed involvement of innate factors such as *C1r-like EGF domain*, *C1q binding protein*, *heat shock protein 70 and 90b*, *galectin*, *membrane attack complex component*/*perforin domain*; *conserved site*, *complement factor D*, *C-type lectin fold*, *il1*, *il1r*, *foxp3*, among others [84]. Another study involving grass carp conducted a transcriptomic and proteomic examination of the intestine after oral DNA vaccination [112]. The study revealed 250 and 50 immune related DEGs and DEPs, respectively, after oral vaccination. KEGG enrichment analysis showed that genes and proteins participating in the Toll-like receptor signaling pathway, MAPK signaling pathway, NOD-like receptor signaling pathway, and the complement cascade were present both in the mucous and tissue homogenates. It is obvious that the innate intestinal mechanisms are quite diverse. More research using omics technologies together with functional assessment will inevitably give us more information about the significance of the various intestinal innate factors that have on innate disease resistance. When it comes to intestinal response after oral vaccination, mucosal response in terms of Ig response is indeed evident [28,113]. Muñoz-Atienza et al. (2021) made a recent overview on mucosal B- and T-cell responses after mucosal vaccination of teleost fish, which we advise the readers to read [2].

### 6.5. Oral Tolerance

Induction of immunological tolerance has been observed in orally vaccinated fish. The cause is probably the activity of regulatory T-cells, as suggested by Marana et al. (2020) [114]. In this study, continuously feeding rainbow trout with a low dosage, *Y. ruckeri* bacterin resulted in lower protection after experimental challenge compared with fish given an initial high dose. Related findings in carp and seabream have been observed [115,116], and in goldfish [25] after feeding protein antigens. It is suggested that primary immunization by oral administration of antigens should be performed using a high initial dose of antigen.

## 7. Conclusions and Future Direction

There are numerous model vaccines that have evoked intestinal immune responses, where Ig response seems to be a major component for protection. However, innate immune mechanisms cannot be ruled out since findings underline that rag^−/−^ zebrafish (without IgM) possess similar protection as rag^−/+^ against experimental *V. anguillarum* challenge. Furthermore, the composition of intestinal microbiota may be changed during oral administration of antigens/plasmids, especially when these are entrapped in vaccine carriers. It is known from studies in higher vertebrates that the gut microbiota may affect vaccine efficacy. Thus, it is pertinent to study whether and how oral vaccination alters intestinal microflora in fish. This also applies to how microbiota affects vaccine efficacy. The maximum efficacy of oral vaccines depends on a prime-boost strategy, how the antigens are protected, how immunogenic the antigens are and the dose. One should not expect higher protection after oral immunization compared to immersion vaccination, indeed and i.p. and i.m. vaccination.

## Figures and Tables

**Figure 1 ijms-22-10932-f001:**
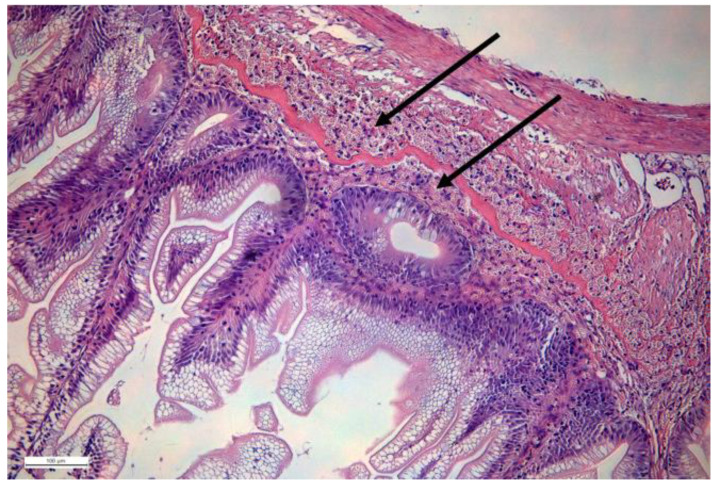
H&E-stained section of salmon intestine. Arrows show localization of EGCs on both sides of the stratum compactum (continuous layer stained red). Courtesy: Lisa Furnesvik and Tore Seternes.

**Table 1 ijms-22-10932-t001:** Overview of recent oral vaccination trials against bacterial diseases.

Disease	Pathogen	Fish Species	Vaccine	Reference
Vibriosis	*V. anguillarum*	Rainbow trout	Anti-*V. anguillarum* antiserum	[31]
		Turbot	Extracellular products	[32]
	*V. alginolyticus*	Silver sea bream	Inactivated	[12]
	*V. anguillarum*	European sea bass	Aquavac Vibrio oral, ISPAH, commercial	[9]
	*V. parahaemolyticus*	Black seabream	DNA in chitosan particles	[11]
	*V. anguillarum*	European sea bass	Commercial Aquavac Vibrio oral	[10]
	*V. harveyi*	Sea bass	Recombinant	[8]
Motile aeromonad septicaemia (MAS)	*A. hydrophila*		LPS+S-layer protein	[33]
		Surubim hybrid	Inactivated bacterin	[13]
		Rohu	Recombinant	[14]
Haemorrhagic septicaemia		*Carp*	*A. hydrophila* ghosts	[34]
Enteric septicemia of carp (ESC)	*E. ictaluri*	Channel & hybrid catfish	Live attenuated	[15]
Edwardsiellosis	*E. tarda*	Olive flounder	Mutated bacteria	[35]
		Rohu	DNA in chitosan	[17]
		Fringed-lipped peninsula carp	Recombinant outer membrane protein A in chitosan	[16]
Yersiniosis	*Y. ruckeri*	Atlantic salmon	*Y. ruckeri* lysate encapsulation in alginate	[18]
		Rainbow trout	Bacterins	[27]
Enteric redmouth disease (ERM)			AquaVac ERM Oral vet (Merck)	[36]
Lactococcosis Streptococcosis	*L. garviae S. iniae*		Eudragit-coated bacteria	[20]
Streptococcosis	*S. agalactiae*	Nile tilapia	Live attenuated/DNA vaccine	[37]
		Red tilapia	Inactivated	[22]
	*S. iniae*	Channel catfish	Recombinant protein in alginate/chitosan	[38]
	*S. agalactiae*	Tilapia	Surface immunogenic protein (SIP) from group B Streptococcus in PMMMA-PLGA	[39]
	*S. iniae*	Red tilapia	Inactivated	[23]
Streptococcosis Motile aeromonad septicemia (MAS)	*S. iniae* *A. hydrophila*	Hybrid tilapia		[24]
Lactococcosis	*L. garviae*	Rainbow trout		[19]
Furunculosis	*A. salmonicida*		Bacterin	[27]
		Turbot	Commercial furunculosis vaccine, Aquavac Furovac	[26]
Salmon rickettsial septicaemia (SRS)	*P. salmonis*	Atlantic salmon	Providean Aquatech 1 Anasac in alginate	[29]
			*P. salmonis* PS2C field strain in MicroMatrix	[40]
Salmonid rickettsial septicaemia Infectious salmon anaemia	*P. salmonis * ISA virus	Atlantic salmon, rainbow trout and coho salmon	Commercial vaccines	[30]
Epizootic ulcerative syndrome	*A. veronii*	Common carp	Recombinant OmpAI	[41]
Enteric septicaemia of catfish	*E. ictaluri*	Channel catfish	Live attenuated *E. ictaluri*	[42]

**Table 2 ijms-22-10932-t002:** Overview of recent oral vaccination trials against viral diseases.

Disease	Pathogen	Fish Species	Vaccine	Reference
Infectious pancreas necrosis	IPNV	Rainbow trout	DNA	[43]
				[45]
				[67]
				[49]
				[46]
		Atlantic salmon	Alginate-encapsulated IPNV antigens	[47]
			Inactivated or live IPNV	[71]
			VP2 DNA vaccine	[48]
		Brown trout & Rainbow trout	DNA	[44]
Nervous necrosis	Red-spotted grouper NNV (RGNNV)	Red-spotted Grouper	Recombinant	[58]
Nervous necrosis	NNV	Orange-spotted Grouper	Subunit	[50]
Viral haemorrhagic disease	Grass carp rheovirus II (GCRVII)	Grass carp	Recombinant	[72]
Spring viremia of carp (SVCV)	SVCV	Common carp		[64]
VHS, IHN	VHS virus, IHN virus	Rainbow trout	Attenuated	[73]
Viral haemorrhagic septicaemia	VHS virus	Olive flounder	Recombinant	[66]
Viral nervous necrosis (VNN)	NNV	Grouper	Inactivated betanodavirus	[53]
Viral nervous necrosis	VNNV		Recombinant	[52]
Viral nervous necrosis (VNN)	VNNV/ Betanodavirus	Seven-band grouper		[58]
Viral nervous necrosis	Piscine nodavirus/Betanodavirus		Inactivated	[59]
Viral nervous necrosis (VNN)	Nodavirus	Orange-spotted grouper	Recombinant	[56]
Viral nervous necrosis (VNN)	Nodavirus	European sea bass	DNA in chitosan	[60]
Viral nervous necrosis (VNN)	Nodavirus	Asian sea bass	DNA in chitosan- tripolyphosphate	[57]
Infectious salmon anaemia	ISAV	Atlantic salmon	Inactivated	[68]
	Rock bream iridovirus (RBIV)	Rock bream	Recombinant major capsid protein	[69]
	Rock bream iridovirus			[70]
Infectious hematopoietic necrosis	IHNV	Rainbow trout	Yeast vaccine EBY 100/pYD1-bi-G	[74]
Grass carp hemorrhagic disease	Grass carp reovirus	Grass carp	Recombinant	[62]

## Data Availability

Not applicable.

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
