# Peer review of "Protection of Teleost Fish against Infectious Diseases through Oral Administration of Vaccines: Update 2021"

_ijms, 2021, doi:10.3390/ijms222010932_

Round 1
Reviewer 1 Report
The Authors of the submitted manuscript titled "Protection of teleost fish against infectious diseases through oral administration of vaccines, an update 2021" are renown aquaculturists, well-established in the field of fish immunology, which makes it no surprise that this paper is highly comprehensive.
I believe that all the important breakthroughs of the last years have been put together in this review, so I do not have any major objections about the content.
I did find, however, some little mistakes in regard to the repetition of Latin species' names or minor stylistic errors. The most important issue which needs to be addressed is to improve the graphical layout of the tables, as indicated below, among the rest of the line-by-line commentary.
- Line 32: Please add the scientific name of the species "(Salmo salar)".
- Line 33: Please add the scientific names of the species: "(Melanogrammus aeglefinus)" and "(Gadus morhua)".
- Line 37: Please add the scientific name of the species "(Oncorhynchus mykiss)".
- Line 45: Please correct "were" to "was".
- Line 62: I would suggest to remove "And" from the start of the sentence and add "also" infront of "showed".
- Line 65: Please correct "were" to "was".
- Lines 74-75: The OmpK gene was already mentioned in Line 54, but only now it was introduced with the full protein name. Therefore, I advise to move this full name to Line 54 and to keep only the abbreviation here.
- Line 77: I believe this sentence was supposed to start as follows: "The OmpK 76 gene expression was found ... ".
- Line 78: The previous comment renders the next iteration of "expression of the ompK gene" unnecessary, improving the style of the 2nd part of the sentence. The whole sentence would sound like this: "The OmpK gene expression was found in mid-intestine, liver, kidney and muscle after administration, and evoked an immune response that protected fish against infection".
- Line 87: Please correct to "sorubim".
- Line 102: Please add the scientific name of the species "(Ictalurus punctatus)".
- Line 110: Please correct to "10.4%".
- Line 119: The scientific name of rohu was already introduced in Line 95.
- Line 171: The scientific name of turbot was already introduced in Line 68. Also please correct to "gilthead sea bream (Sparus aurata)".
- Line 179: Please correct to "lymphoid".
- Line 187: Please italicize "niloticus".
- Line 195: Please add the scientific name of the species "(Oncorhynchus kisutch)".
- Line 200: Please add the missing parenthesis opening of "(2020)".
- Table 1: I have several remarks here, as this table appears to be hastily formatted:
- I would suggest to proceed with a graphical upgrade by adjusting column width - the last "Reference" column can be as narrow as its title, leaving more space for the other columns, so that the number of unnecessary line breaking is reduced, especially in the "Pathogen" column.
- Besides, I would suggest to merge the table cells in each column if they contain the same phrases, thus grouping some of the lines and avoiding repetition. For instance, looking just at the part shown on Page 5, there would be:
- one cell in the first column with a single "Vibriosis";
- then, there would be four cells with pathogens (V. alginolyticus, V. anguillarum, V. parahaemolyticus, V. harveyi).
- then, in the "Fish species" column, there could possibly be one merger, as there were two studies on D. labrax regarding V. anguillarum infection ([9] and [10]).
- The last two columns would likely remain without any merging throughout the whole table.
- I would also suggest to either include the scientific names of species consistently in each cell, or to remove them completely.
- The latter option above would further allow to reduce the number of unnecessary double-lined cells, making the whole table more clear to read. I think that using a smaller font would also help to remove most of the double- or triple-spacing in columns 2-4. The table in its current form stretches almost over 4 whole pages, which is quite excessive.
- Furthermore, I also advise to use horizontal lines. To illustrate the final layout, I have attached a file with an exempliary beggining of this Table.
- Line 247: Please avoid starting the sentence with "And".
- Line 260: The scientific name of European sea bass was already introduced in Line 58.
- Line 262: The scientific name of turbot was already introduced in Line 68.
- Line 263: The scientific name of Atlantic cod was already introduced in Line 33 (after corrections).
- Line 290: The scientific name of European sea bass was already introduced in Line 58.
- Line 325: There is no need to introduce the full "immunoglobulin M" here, "IgM" has already appeared many times before.
- Line 356: Please correct to "IHNV G".
- Line 363: The scientific name of Atlantic salmon was already introduced in Line 32 (after corrections).
- Line 366: The scientific name of brown trout (sea trout) was already introduced in Line 228.
- Table 2: I have the exactly same objections in terms of formatting as for Table 1. Please improve the layout accordingly.
- Lines 387-388: The scientific name of grass carp was already introduced in Line 298. In fact, I believe that this whole sentence belongs there as well, when the "grass carp industry" is mentioned.
- Line 407: The scientific name of gilthead sea bream was already introduced in Line 171.
- Table 3: I have the exactly same objections in terms of formatting as for Tables 1 and 2. Please improve the layout accordingly.
- Line 466: Please correct "is" to "as".
- Line 488: Please italicize "S. agalactiae".
- Line 520: Please correct to "known".

Author Response
We are very pleased with the thorough review done. As such, the revised manuscript will improved significantly. As can be seen from the revised manuscript, we have point-for-point addressed all comments made (followed the referee´s suggestion). Regarding the issue of formatting tables/layout: We have reformatted the tables to ease reading, left out the fishes´ latin names, and merged cells throughout. The production assistants may center the table more in line with the dpi standards, and add lines below cells. We have expanded the chapter on vaccines against parasites (cf. referee 2). The production assistant may also help to format the references, which is not very well displayed from my computer.
Reviewer 2 Report
Comments
Manuscript ijms-1402212 is an interesting review paper highlighting the advances of oral administration of vaccines for evoking the protection of teleost fish against infectious diseases. After carefully reading, some major and minor issues are found in the ms that need to be addressed before the ms is fit for publication. Many redactional errors are found especially in term of writing the name of species. Section of parasitic diseases needs to be improved since it has poor information, as we known that a lot of related reports have existed.
Introduction
29 - should be “zebrafish (Danio rerio)
32 - Atlantic salmon (Salmo salar)
33 - haddock () and cod (Gadus morhua)
- …..12 mg ml-1 Igs, respectively.
37 - rainbow trout (Oncorhynchus mykiss)
Bacterial diseases
51 - V. harveyi
55 - V. anguillarum
62 - should be “….., and fish treated…..”
65 - was
76-77 - Should be “The ompK gene transcripts were found…..”
80 - V. alginolyticus
90 - A. hydrophila
98 - …..the relative percent survival (RPS).
101 - include the scientific name of fish species!
117 - E. tarda
118 - should be “rohu”
119 - “(L. rohita)” can be omitted!
121-2 - I would recommend the authors to follow the way writing the name of gene in zebrafish!
156 - A. hydrophila
171 - “(Scophthalmus maximus)” can be omitted!
176 - A. salmonicida
178 - Y. ruckeri
187 - niloticus
194 - gram-negative bacterium
195 - mention the scientific name for coho salmon!
210 - The authors should mention “Table 1” in the text.
- Lines for row and column are missing in the Table.
- The name of fish species should be written consistently. If it comes with the scientific name, then apply it for all of the species.
Viral diseases
223-4 - “relative percent survivals” can be omitted!
227 - RPS of 80%
229 - showed
231 - ……, but …..
247 - ….., and ….
254-5 - see the previous comment in lines 121-2!
260-1 - “(Dicentrarchus labrax)” can be omitted!
262 - “(Scophthalmus maximus)” can be omitted!
265 - V. anguillarum
267 - see the previous comment in lines 121-2!
290 - “(Dicentrarchus labrax)” can be omitted!
307 - RPS rates of
308 - “(C. Idella)” can be replaced with “grass carp”
328 - L. plantarum
336 - correct the font size!
363 - “(Salmo salar L.)” can be omitted! See the previous comment in line 32!
372-4 - modify these sentences!
382 - see the previous comment regarding the Table! (line 210)
Parasitic diseases
383 - this section has poor information regarding the oral vaccination in combating parasitic disease as the authors are only talking about one kind of parasitic disease Clonorchiasis. There are many parasites out there affecting fish and aquaculture worldwide. Oral vaccination has been being applied for some of those parasitic diseases. For example: parasite Ichthyophthirius multifiliis is an important parasite which infecting almost all freshwater fish species. Vaccination including orally administrated has been tried to cope this diseases as reported by Yao et al 2016 (https://pubmed.ncbi.nlm.nih.gov/27663853/) and others. Mutoloki et al 2015 (https://www.ncbi.nlm.nih.gov/pmc/articles/PMC4610203/) has also reported the oral vaccination against Sea lice. The authors can improve this section by including these mentioned reports and others.
387 - “(Ctenopharyngodon Idella)” can be omitted!
Model antigens
404 - “, Danio rerio” can be omitted!
407 - “(Sparus auratus)” can be omitted!
418 - see the previous comment regarding the Table! (lines 210 & 382)
Intestinal immune defense
441 - see the previous comment in lines 121-2!
443, 450, 451, 470 - zebrafish
473 - Figure 1 should be mentioned in the text!
507 - Y. ruckeri
Conclusion and future direction
I would recommend modifying the conclusion by omitting the references as they are not necessary to be included.

Author Response
We are very pleased with the very good and detailed review done. The revised manuscript will be significantly improved. As can be seen from the revised manuscript, we have point-for-point addressed all comments made (followed the referee´s suggestion). Regarding the issue of formatting tables/layout: We have reformatted the tables to ease reading, left out the fishes´ latin names, and merged cells throughout. The production assistants may center the table more in line with the mdpi standards, and add lines below cells. We have expanded the chapter on vaccines against parasites. The production assistant may also help to format the reference section, which is not very well displayed from my computer.
Round 2
Reviewer 2 Report
Comments
Manuscript ijms-1402212 has been significantly improved by the authors based on previous comments, but some of items are not fully addressed and even not responded. I would strongly recommend addressing the existing issues found in order to make the ms is fit for publication.
124 - should be “ighc, inos, tlr22, nod1, and il-1β”. As mentioned previously, the authors can follow the way of writing the gene name in zebrafish. See https://zfin.org/search?category=Gene+%2F+Transcript&q=
256-7 - see the previous comment regarding gene name in line 124!
268 - see the previous comment in line 124!
300 - should be “grass carp (Ctenopharyngodon idella)”
310 - “(Ctenopharyngodon idella)” can be omitted!
413 - see the previous comment in line 124!
415 - I. multifiliis
417 - I. multifiliis
463 - see the previous comment in line 124!
495 - mention Figure 1 in the text!
535- As mentioned previously, modify the conclusion by omitting the references as they are not necessary to be included!

Author Response
Thanks for the repeatedly notice that zebrafish genes must be in lower case and in italics. We are sorry that we missed such details (and important) from the first round of comments - made by the referee 2. We hope that we have managed to include all details now. Please see MS for the second revision.